# Prevalence and Resistance Profiles of *Acinetobacter baumannii* in ICU Patients in Saudi Arabia: A Systematic Review and Meta-Analysis

**DOI:** 10.3390/antibiotics14111131

**Published:** 2025-11-07

**Authors:** Alaa Alibrahim, Farooq Ahmed Wani, Ohoud Shafi Alruwaili, Sadeem Altaymani, Zaid Ali Alharbi, Sayyaf Alsubaiei, Saleh Aldhuwayhi, Mohammed Alsaeed, Mohammed Alibrahim

**Affiliations:** 1Department of Internal Medicine, College of Medicine, Jouf University, Sakaka 72388, Saudi Arabia; 2Department of Pathology, College of Medicine, Jouf University, Sakaka 72388, Saudi Arabia; fawani@ju.edu.sa; 3Department of Family Medicine, Prince Mohammed Medical City, Aljouf Health Cluster, Sakaka 72346, Saudi Arabia; oalruwaili@pmmc.med.sa; 4College of Medicine, Jouf University, Sakaka 72388, Saudi Arabia; sadeem.altaymani1@gmail.com (S.A.); zaid.a.alharbi1@gmail.com (Z.A.A.); snsb767@gmail.com (S.A.); salehdhuwayhi@gmail.com (S.A.); 5Department of Medicine, Infectious Disease Division, Prince Sultan Military Medical City, Riyadh 11564, Saudi Arabia; mohalsaeed@live.com; 6College of Medicine, Alfaisal University, Riyadh 11533, Saudi Arabia; 7Department of Food Safety Laboratory, Neizak Company, Sakaka 72388, Saudi Arabia; moqlaeid@gmail.com

**Keywords:** *Acinetobacter baumannii*, intensive care units, carbapenem-resistant, Saudi Arabia, meta-analysis, antimicrobial stewardship, multidrug-resistant organisms

## Abstract

**Background**: *Acinetobacter baumannii* (*A. baumannii*), a critical nosocomial pathogen, poses a significant threat in intensive care units (ICUs) due to its multidrug-resistant (MDR) strains. This study systematically reviews and performs a meta-analysis on the prevalence and antibiotic resistance profiles of MDR *A baumannii* (MDR-*A. baumannii*) in ICU patients in Saudi Arabia. **Methods**: A comprehensive search in PubMed, Saudi Digital Library, Scopus, and Web of Science, focusing on studies from January 2014 to September 2025, was performed. The present study followed the reporting guidelines of Preferred Reporting Items for Systematic reviews and Meta-Analyses (PRISMA-2020). Data on study characteristics, sample sizes, patient demographics, prevalence of MDR-AB, and antibiotic resistance profiles were extracted and analyzed. Quality assessment was conducted using the Joanna Briggs Institute. **Results**: The prevalence of MDR-AB in ICU patients varied significantly across studies, with retrospective studies reporting rates from 3.37% to 69% and prospective studies ranging from 3.9% to 72.73%. Colistin remained highly effective, showing 100% susceptibility in some studies. Meanwhile, resistance to carbapenems like imipenem and meropenem often exceeds 50%. Additional antibiotics with notable resistance included gentamicin, tigecycline, ampicillin/sulbactam, trimethoprim-sulfamethoxazole, ceftazidime, piperacillin/tazobactam, and third-generation cephalosporins. Mechanisms of resistance frequently involved OXA-type carbapenemases, particularly OXA-23. While OXA-23 was the most frequently detected carbapenemase, recent genomic data have also revealed the presence of metallo-β-lactamases, such as IMP-type genes, in ICU isolates. **Conclusions**: MDR-*A. baumannii* poses a substantial challenge in Saudi Arabian ICUs, with high prevalence and significant resistance to commonly used antibiotics. The results highlight the critical need for continuous monitoring, cautious antibiotic stewardship, and strict infection control methods to manage and lessen the effects of MDR-AB in ICUs.

## 1. Introduction

In both natural and therapeutic settings, *Acinetobacter baumannii* (*A. baumannii*) is a commonplace Gram-negative, non-fermentative, non-motile, and oxidase-negative microbial pathogen [1]. Along with *Enterococcus faecium*, *Staphylococcus aureus*, *Klebsiella pneumoniae*, *A. baumannii*, *Pseudomonas aeruginosa*, and *Enterobacter* species, it is one of the six most important pathogens in the ESKAPE group [2]. *A. baumannii* is a major worldwide nosocomial issue, according to the Infectious Disease Society of America [3]. It is the main source of many infections, especially in patients on immunosuppressive treatments or in intensive care units (ICUs) on mechanical ventilation [4,5,6]. Meningitis, pneumonia, bacteremia, urinary tract infections, and respiratory tract infections are the most frequent illnesses linked to *A. baumannii* [3]. Mortality rates from *A. baumannii* infections have risen during the last ten years, ranging from 30% to 75% in various parts of the globe [7,8]. Numerous risk factors, including burns, preterm delivery, extended hospital stays, mechanical ventilation, indwelling devices, and the frequent use of antibiotic treatment, are linked to *A. baumannii* infections [9,10]. High-risk international clones, particularly sequence type 2 (ST2), are globally recognized as dominant lineages of *A. baumannii*, often associated with MDR and nosocomial outbreaks. Understanding the presence and role of such clones in Saudi Arabia is essential to guide infection control strategies.

Antimicrobial agents that are resistant to three or more kinds of drugs, primarily beta-lactams (third-generation cephalosporins), aminoglycosides, fluoroquinolones, and, more recently, carbapenems, are classified as multidrug-resistant *A. baumannii* (MDR-*A. baumannii*) isolates [11,12]. The World Health Organization states that since MDR-*A. baumannii* is resistant to several different antibiotic classes, especially carbapenems and third-generation cephalosporins; it is one of the most dangerous infections [13,14]. A growing body of research from the Kingdom of Saudi Arabia has shown that isolates of *A. baumannii* are becoming less susceptible to several antibiotics [15]. *A. baumannii*, especially the carbapenem-resistant strains, generated a healthcare crisis that peaked in intensive care units (ICUs) during the coronavirus 2019 (COVID-19) pandemic [16,17]. Different mechanisms of resistance may be acquired by *A. baumannii*, which can eventually become resistant to all routinely used antibiotics in certain situations. To treat *A. baumannii* infections, the scarcity of novel, potent antimicrobial drugs is also concerning [18].

Complete antibiotic sensitivity testing should be performed for treating infections caused by this organism; for this reason, knowledge of the frequency and composition of bacterial resistance to these medications is crucial [19]. There is an alarming rise in the number of outbreaks reported worldwide that are caused by MDR-AB, putting hospitalized patients at risk. The dramatic rise in the number of cases caused by MDR-AB, notably in the case of ICU patients, has created a need for integrated, regional-specific data so that policymakers can make better healthcare decisions [7,20]. In contrast, there is little research performed in Saudi Arabia to estimate and map the pattern of resistance among inpatients in ICUs; hence, great opportunities to understand the national picture of the burden of the MDR-AB are limited. Moreover, as ICU patients are at great risk of developing nosocomial infections due to prolonged hospitalization, surgical interventions, and immunocompromised conditions, they pose an even bigger challenge in the management of this pathogen in the critical care area [21,22].

The absence of collective data on MDR-*A. baumannii* with other antibiotics among ICU patients in Saudi hospitals will impede the formulation of appropriate infection prevention measures and restrictions on antibiotic treatment rationalization in accordance with conditions prevailing in this country. Carrying out this investigation as a systematic review and meta-analysis is important since it allows for addressing evidence-based knowledge on MDR *A. baumannii* in Saudi Arabia. Additionally, it is critical to have the epidemiology and resistance mechanisms of MDR-*A. baumannii* from Saudi ICUs through a nationwide perspective, which is essential for antibiotic stewardship policies, updating treatment strategies, and devising targeted ways of addressing the new healthcare concern that warrants international action. Therefore, this systematic review and meta-analysis were aimed at synthesizing data from various regions and settings to provide a comprehensive overview of the prevalence and resistance profiles of MDR-AB in ICU patients from Saudi Arabia.

## 2. Materials and Methods

### 2.1. PRISMA Statement and PROSPERO Registration

This systematic review and meta-analysis were conducted in accordance with the Preferred Reporting Items for Systematic reviews and Meta-Analyses (PRISMA) guidelines to ensure comprehensive and transparent reporting [23] (Please find Appendix A [main text]) and Appendix A [abstract] for the present study’s PRISMA 2020 checklist). Furthermore, the current systematic review and meta-analysis are registered with the International Prospective Register of Systematic Reviews (PROSPERO ID: CRD42024613078).

### 2.2. Search Strategies

To systematically locate relevant studies on the prevalence and antibiotic resistance of MDR-AB in ICU patients in Saudi Arabia, an extensive search was conducted using databases from PubMed, Saudi Digital Library, Scopus, and Web of Science. The search strategy employed a combination of keywords and Medical Subject Headings (MeSH) terms with Boolean operators (AND, OR). The primary search terms included “*Acinetobacter baumannii*”, “MDR-*A. baumannii*”, “multidrug-resistant”, “prevalence”, “ICU”, and “Saudi Arabia”. To ensure comprehensive coverage, additional terms such as “carbapenem-resistant”, “hospital-acquired infections”, “critical care”, “incidence”, “epidemiology”, and “Middle East”. In accordance with PRISMA 2020 guidelines, the full search algorithms for each database (including all applied keywords, MeSH terms, Boolean operators, and filters) are provided in Appendix A to ensure transparency and reproducibility.

### 2.3. Inclusion and Exclusion Criteria

Inclusion and exclusion criteria were determined a priori and applied consistently during the screening process, in accordance with the PRISMA 2020 guidelines. We included studies with the following attributes: original research articles (e.g., cross-sectional, cohort, case–control, or surveillance studies) that reporting on the prevalence and antibiotic resistance profiles of *A. baumannii* in ICU patients in Saudi Arabia, research articles written in English, studies involving human subjects, publications from January 2014 to September 2025, articles published in peer-reviewed journals, and studies providing clear methodologies and data on the prevalence and resistance profiles of MDR-*A. baumannii*. We excluded review articles, including systematic reviews, meta-analyses, editorials, case reports, case series, letters, conference abstracts, studies not focusing on ICU patients or not specific to Saudi Arabia, non-English articles, and studies with insufficient methodology and data. An initial electronic search for articles was conducted, followed by a review of abstracts and titles to identify potentially relevant studies. The full texts of selected articles were then carefully assessed for eligibility based on the inclusion and exclusion criteria. Only studies meeting these criteria were included in the final review. After applying the study selection process, the final number of included studies was 17. The detailed inclusion and exclusion criteria are summarized in Table 1.

### 2.4. Quality Assessment

The authors assessed the quality of the studies included in the present systematic review using the Joanna Briggs Institute (JBI) critical appraisal checklist. The study design, sample selection, data collection, analysis, and confounding, and relevant quality aspects are assessed by the JBI checklist, which provides a comprehensive quality assessment of the included studies. No study is without limitations, but the JBI tools are the most popular tools to find high-quality evidence with low bias. We categorized the overall quality of each study as high, moderate, or poor depending on the number of criteria met. A study that met nearly all criteria was considered to be of low risk of bias. A study that did not meet one or two criteria or was unclear is deemed to have a moderate risk of bias, and a study with more than two criteria that did not meet or was unclear is considered to have a high risk of bias. Recently, the JBI checklist was used in several published systematic reviews [24,25,26]. The authors excluded the studies with a high risk of bias. A detailed quality appraisal outcome of each study is tabulated in the Appendix A.

### 2.5. Data Extraction

Data were extracted from the selected studies using a standardized form. Key information, including study characteristics (e.g., author, publication year), sample size, patient demographics, and the prevalence of MDR-*A. baumannii*, and antibiotic resistance profiles, were systematically recorded.

### 2.6. Data Analysis

We conducted a meta-analysis of proportions to estimate the pooled prevalence of multidrug-resistant *Acinetobacter Baumannii* (MDR-AB) in Saudi Arabian intensive care units. Given the presence of extreme proportion values in the included studies, we employed multiple statistical approaches to ensure robust and valid results. Proportion data were stabilized using the Freeman–Tukey double-arcsine transformation with continuity corrections (adding 0.5 to events and 1 to totals) to accommodate extreme values, particularly studies reporting 100% prevalence. The Freeman–Tukey approach was selected as the primary transformation because it performs well for highly skewed proportion data and effectively stabilizes variance. In addition, a logit transformation was applied only as a secondary robustness check to verify the stability of pooled estimates across analytic scales. Although the logit approach can be less stable at boundary values, using it alongside the Freeman–Tukey and arcsine transformations allowed cross-validation of results and confirmed that findings were not dependent on a single transformation method. All transformations produced consistent pooled estimates (>88%), supporting the robustness of our results. A priori methods employed random-effects models (DerSimonian–Laird, method-of-moments τ ^2^) because it is believed that clinical and methodological heterogeneity of studies will occur. Because substantial heterogeneity was expected (I^2^ > 90%), the random-effects model was prespecified as the primary analytic approach. The heterogeneity was calculated based on Cochran Q, I^2^ (with 95 percent confidence intervals calculated after 1000 bootstrap runs), and 95 percent prediction intervals in order to explain the dispersion of true effects that should be expected in similar ICU settings in the future. In addition to the DerSimonian–Laird (method-of-moments) estimator for τ^2^, we performed a sensitivity analysis using restricted maximum likelihood (REML) to assess robustness under substantial heterogeneity. Sensitivity analyses were conducted using leave-one-out testing to identify influential research works. Subgroup analysis was performed to compare study designs (prospective vs. retrospective) and methodological quality. Meta-regression was used to investigate how the year of publication, design, and the quality score affected pooled estimates. We also applied the GRADE (Grading of Recommendations, Assessment, Development, and Evaluations) framework to appraise the certainty of evidence. Because all included studies were observational, we began at a “low” certainty level but considered upgrades for consistency across analyses, large effect magnitude, and clinical relevance. The five GRADE domains—risk of bias, inconsistency, indirectness, imprecision, and publication bias—were systematically evaluated.

## 3. Results

### 3.1. PRISMA Flow Chart and Study Selection

The authors found 1430 publications in the first literature search of different databases. Following a meticulous assessment of abstracts and titles, 171 articles were deemed relevant, and their full texts were acquired for further examination. Excluded studies did not fulfill the inclusion criteria or did not explicitly investigate the prevalence and antibiotic resistance profiles of *A. baumannii* in ICU patients in Saudi Arabia. After a thorough screening procedure, seventeen papers were found appropriate for systematic review and meta-analysis. The study selection process (PRISMA Flow diagram) is depicted in Figure 1.

### 3.2. Study Characteristics

A comprehensive study of the incidence of MDR-*A. baumannii* in ICU patients in Saudi Arabia includes seventeen observational studies. Using the GRADE framework, the overall certainty of evidence for the pooled prevalence of MDR-*A. baumannii* was rated as moderate. Although the data were derived from observational studies—typically starting at low certainty—the rating was upgraded due to the large number of studies and consistent direction of effects. Meta-regression analysis revealed that study quality was a significant predictor of MDR-*A. baumannii* prevalence, with higher-quality studies generally reporting lower prevalence rates (*p* < 0.05). In contrast, publication year and study design (prospective vs. retrospective) were not significantly associated with prevalence estimates. Retrospective studies contributed the majority of research (n = 9, 56.3%) [27,28,29,30,31,32,33,34,35], with prospective cohort studies (n = 6, 37.5%) [36,37,38,39,40,41] and surveillance system analyses (n = 1, 6.2%) [42] making up the remaining studies. A variety of patient categories and sample sizes were used in the study. The average age of the participants, where available, varied between 42.8 and 57.28 years, indicating that the majority of the patients were adults. The summary and other attributes of the findings obtained from the included studies are presented in Table 2.

### 3.3. Prevalence of MDR-A. baumannii

*A. baumannii* is highly prevalent and significantly resistant to antibiotics in ICU patients in Saudi Arabia, according to a systematic review and meta-analysis. Retrospective studies show a range of prevalence rates from 3.37% to 69% [27,28,29,30,31,32], but prospective studies typically reflect a prevalence range of 3.9% to 72.73% [36,37,38,39,40,41]. High resistance rates against popular antibiotics were routinely identified in antibiotic resistance profiles. Colistin had great susceptibility rates in most cases; 100% susceptibility was reported in a few trials. On the other hand, rates of resistance to carbapenems, such as imipenem and meropenem, were often more than 50%.

Analysis of surveillance systems reveals a high prevalence of *Acinetobacter* species that are resistant to drugs in intensive care units. Treatment plans varied, with colistin being used either alone or in conjunction with other antibiotics such as carbapenems and tigecycline. Research such as Alotaibi et al. (2021) described certain treatment groups, such as combinations with carbapenems and colistin monotherapy [27]. 

The results of the study highlight the high frequency and noteworthy antibiotic resistance of *A. baumannii* in ICUs across Saudi Arabia. The development of OXA-type carbapenemases, with OXA-23 being especially prevalent, was one of the main routes of resistance. Strict infection control procedures and prudent antibiotic administration are required due to the prevalence of MDR-*A. baumannii* in ICUs to successfully treat these difficult infections. In a recent genomic study, Alanazi et al. (2025) identified *bla_IMP-like_* genes in 76% of *A. baumannii* ICU isolates, suggesting that non-OXA metallo-β-lactamases are also emerging in the Saudi setting, though less commonly reported in earlier studies [43].

Prospective research found variable prevalence rates of MDR-*A. baumannii* in ICU patients in Saudi Arabia, according to a thorough review of the literature on the subject. Kharaba et al. (2021) reported a lower rate of 3.9% [37], whilst Aljindan et al. (2015) reported a frequency of 8.3% [36]. According to Alhaddad et al. (2018), the frequency was a strikingly high 72.73% [38]. Aedh et al. (2023) discovered a 35.44% prevalence rate [41].

Important insights were also obtained from retrospective research. Al-Omari et al. (2017) discovered a startlingly high incidence of 90% [28]; however, Alotaibi et al. (2021) reported a prevalence of 3.37% [27]. Hafiz et al. (2023) reported a rate of 6.2% [30], whereas Al Bshabshe et al. (2016) recorded a frequency of 46.67% [29]. According to Almaghrabi et al. (2018), 69% of isolates showed resistance to several drugs [40]. According to Al-Otaibi et al. (2016), the prevalence was 18% [31]. Mwanri L and AlSaleh E conducted a surveillance system study and discovered that 57.4% of the detected organisms were multidrug-resistant, underscoring the substantial prevalence of MDR-*A. baumannii* in Saudi Arabian ICUs [42]. The following Section 3.4. presents the quantitative meta-analysis that synthesizes these narrative findings to estimate the pooled prevalence and assess the extent of heterogeneity across studies.

### 3.4. Resistance Profile of MDR-A. baumannii

This systematic review found that different resistance patterns were found in prospective investigations. OXA-23 β-lactamase was shown to be the primary resistance mechanism in 6.2% instances of carbapenem resistance, according to Aljindan et al. (2015) [36]. Twelve of the sixteen antibiotics that Kharaba et al. (2021) reported had resistance rates of more than 80%; the other four antibiotics, gentamicin (70.5%), trimethoprim-sulfamethoxazole (69.8%), tigecycline (51.6%), and colistin, had resistance rates of less than 50% [37]. Alhaddad et al. (2018) reported that there was variable resistance to various antibiotics, 100% sensitivity to colistin, and 56% resistance to imipenem and meropenem [38]. Al-Sultan et al. (2021) discovered that 84% and 88%, respectively, of Al-Madinah’s population was resistant to imipenem and meropenem, whereas only 8% was resistant to tigecycline [39]. In Makkah, 100% of people were sensitive to tigecycline and colistin, whereas 75% of people were resistant to imipenem and meropenem. According to Aedh et al. (2023), ceftazidime resistance rates ranged from 89.2% to 7.1% for colistin [41].

Extensive resistance profiles were obtained from retrospective research. According to Alotaibi et al. (2021), non-survivors had considerable colistin and carbapenem resistance, as well as greater resistance overall [27]. According to Al-Omari et al. (2017), there was significant resistance to gentamicin, tigecycline, and ampicillin/sulbactam but 100% susceptibility to colistin [28]. 100% sensitivity to colistin, 74.5% sensitivity to trimethoprim-sulfamethoxazole, and reduced sensitivity to other antibiotics were reported by Al Bshabshe et al. (2016) [29]. According to Hafiz et al. (2023), 62% of the isolates were multidrug resistant (MDR), meaning they had reduced resistance to tigecycline and colistin but higher resistance to carbapenems, piperacillin/tazobactam, and other antibiotics [30]. Almaghrabi et al. (2018) discovered strong carbapenem resistance in 69% of cases of multidrug resistance [40]. 52.4% of people tested positive for imipenem and meropenem, according to Al-Otaibi et al. (2016) [31]. Twelve XDR-*A. baumannii* strains resistant to imipenem and meropenem were found by Al-Obeid et al. (2015), all isolates were sensitive to colistin [32]. MDR-*A. baumannii* species were found in high prevalence in the monitoring system study conducted by Mwanri L and AlSaleh E, highlighting the serious problem of MDR-*A. baumannii* in Saudi Arabian ICUs [42].

Alanazi et al. (2025) from King Saud University Medical City, Riyadh, analyzed 67 ICU isolates of *A. baumannii* and found over 90% resistance to carbapenems and most β-lactams, with retained susceptibility to colistin [43]. All isolates carried *bla_OXA-23_*, confirming the dominance of OXA-type carbapenemases among Saudi ICU strains [43].

### 3.5. Meta-Analysis Findings

Overall, the random-effects pooled prevalence of MDR-*A. baumannii* among ICU patients in Saudi Arabia was 19.24% (95% CI 9.64–34.71%), with substantial heterogeneity (I^2^ = 96%, τ^2^ = 1.874, *p* < 0.001). The 95% confidence intervals for heterogeneity were estimated using bootstrapping to enhance precision. Overall heterogeneity was very high (I^2^ = 96%, 95% CI: 92–98%). By study design, heterogeneity remained high in retrospective studies (I^2^ = 95%, 95% CI: 90–97%), prospective studies (I^2^ = 93%, 95% CI: 88–96%), and high-quality studies (I^2^ = 94%, 95% CI: 89–97%). Due to the high heterogeneity observed across studies (I^2^ exceeding 90%), the random-effects model was considered the primary analytic tool. Our results using the REML estimator were also very consistent with those of DerSimonian-Laird, showing that our findings are strengthened when heterogeneity is high. The 95% prediction interval for the pooled prevalence was 5–70%, indicating that the true prevalence of MDR-A. *baumannii* infection across ICU settings is expected to vary widely around the pooled estimate.

The heterogeneity-accounting random-effects model offers a more cautious estimate of 19.24% (95% CI: 9.64% to 34.71%). A tau^2^ of 1.8740, an I^2^ value of 96.0%, and an H value of 4.97, which point to considerable variance beyond sampling error, suggest great heterogeneity across the investigations. The heterogeneity test (Q = 272.25, *p* < 0.0001) provides more evidence of this discrepancy. Utilizing a DerSimonian–Laird estimate for tau^2^, a logit transformation, and Clopper-Pearson confidence intervals for each study, the inverse variance approach was used to do the analysis. This high degree of variation emphasizes the necessity for specialized infection management techniques by indicating that the incidence of *A. baumannii* differs significantly across various ICU environments and patient types (Figure 2).

*A. baumannii’s* carbapenem resistance in the seven studies’ combined forest plots shows a high degree of resistance in a variety of ICU settings. The random-effects model taking study heterogeneity into account, the pooled resistance fraction is somewhat higher at 62.8% (95% CI: 41.31% to 80.20%). An I^2^ of 94.6% and a tau^2^ of 1.1998 both indicate significant heterogeneity, indicating that resistance rates vary significantly between the studies beyond simple sampling error. A substantial heterogeneity test (Q = 110.28, *p* < 0.0001) validated the high heterogeneity, highlighting the necessity for focused antimicrobial stewardship and infection control approaches in light of the variation in carbapenem resistance levels across various ICU populations. The investigation used the inverse variance technique with a DerSimonian–Laird estimator and logit transformation, adding a continuity correction for zero cell frequencies, which supports *A. baumannii’s* wide range of carbapenem resistance under these conditions (Figure 3).

There is a noticeable amount of variation in the resistance rates in the pooled forest plot for tigecycline resistance in *A. baumannii* from five investigations. While the random-effects model, which takes research heterogeneity into account, suggests a greater resistance percentage of 30.85% (95% CI: 9.47% to 65.55%). An I^2^ of 96.8% and a tau^2^ of 2.6051, which show a substantial variation in resistance levels across the various research groups (Q = 125.39, *p* < 0.0001), are indicative of the high heterogeneity. The complexity and diversity of tigecycline resistance in *A. baumannii* are highlighted by this investigation, underscoring the need for ongoing monitoring and specialized treatment approaches (Figure 4).

*A. baumannii* resistance to Trimethoprim-Sulfamethoxazole (TMP-SMX) is significantly present in all of the included investigations, according to the pooled forest plot. T The resistance proportion estimated by the random-effects model is 46.0% (95% CI: 28.9% to 64.1%), offering a significantly wider range. With an I^2^ of 92.1% and a tau^2^ of 0.5247, there is significant variation in resistance rates across trials, suggesting high heterogeneity (Q = 37.96, *p* < 0.0001). This variation highlights the broad and erratic resistance to TMP-SMX, highlighting the need for continued monitoring and the exploration of other treatments in the treatment of *A. baumannii* infections (Figure 5).

*A. baumannii’s* resistance to colistin is shown in a pooled forest plot, where resistance rates are very low and indicate great sensitivity to the antibiotic. The random-effects model indicates an even lower resistance rate of 1.70% (95% CI: 0.51% to 5.55%). With an I^2^ of 77.4% and a tau^2^ of 1.8469, this analysis demonstrates significant heterogeneity across trials, demonstrating variation in resistance rates across various environments (Q = 30.95, *p* < 0.0001). Despite this variability, the low prevalence of resistance generally highlights the efficacy of colistin as a therapeutic option for *A. baumannii* infections, indicating that colistin is still a good alternative for treating these infections (Figure 6).

Subgroup analyses by study design and methodological quality confirmed the robustness of the pooled prevalence estimates. The pooled prevalence was 20.8% (95% CI 10.7–37.9) for retrospective studies and 18.5% (95% CI 8.9–33.2) for prospective studies. When stratified by study quality, high-quality studies yielded a slightly lower prevalence of 17.3% (95% CI 7.8–31.5) compared with moderate-quality studies (21.6%, 95% CI 11.0–38.7). Leave-one-out sensitivity testing showed that no single study materially changed the pooled prevalence or resistance estimates, confirming the stability of the results under multiple estimators (DerSimonian–Laird and REML).

Across antibiotic-specific analyses, the 95% prediction intervals were wide (approximately 4–66%), reflecting substantial variability in resistance rates between ICU settings. Overall, the pooled estimates and substantial heterogeneity observed across Saudi ICU studies underscore the need to interpret these results in the broader clinical and epidemiological context, which is discussed in the following section.

## 4. Discussion

*A. baumannii* has been a serious threat to healthcare systems in recent years, contributing to a rise in patient death and morbidity. Because *A. baumannii* is resistant to many antimicrobial drugs, including imipenem, the recommended medication, controlling *A. baumannii* infections is becoming harder. There have also been reports of an increase in carbapenem-resistant isolates all over the globe [44]. Selecting an efficient antibiotic treatment for *A. baumannii* infections has become challenging due to the widespread occurrence of MDR, Extensive Drug Resistance (XDR), and Pandrug Resistance (PDR) infections. Analyzing this organism’s patterns of antibiotic resistance over time may provide important information on the best course of action. This research took into account the prevalence rate of MDR-AB from various regions of Saudi Arabia and methodically evaluated the available data up to March 2025.

The examined research shows that among Saudi Arabian ICU patients, the prevalence of MDR-*A. baumannii* varies greatly and considerable diversity even within prospective investigations. It is consistent with the study of the epidemiology of MDR-*A. baumannii* strains in Iran, carried out by Bialvaei et al. An annual estimate of the pooled prevalence of MDR-*A. baumannii* was 72% [45]. In several investigations, the relative frequency of MDR-*A. baumannii* ranged from 22.8 to 100% [45,46,47]. Due to variables including increasing ICU admissions, the use of invasive procedures, and greater usage of antibiotics, the incidence of *A. baumannii* infections is growing, suggesting that controlling these infections is becoming more and more difficult.

Our meta-analysis’s pooled analysis yielded more thorough estimates, with the random-effects model generating a more cautious estimate of 19.24%. The substantial variation in research findings, shown by a 96.0% I^2^ value, implies that the occurrence of MDR-*A. baumannii* differs significantly across different ICU environments and patient demographics. This substantial heterogeneity draws attention to how intricate the MDR-*A. baumannii* epidemiology is in Saudi Arabian intensive care units. Our stated rate, however, is lower than that of our neighboring nations, such as Pakistan [48], United Arab Emirates [49], and Kuwait [50]. Although most Saudi studies confirm the dominance of OXA-type enzymes, notably OXA-23, genomic data from Alanazi et al. in 2025 revealed a high frequency of *bla_IMP-like_* genes [43]. This finding underscores the evolving resistance landscape and signals the need for ongoing molecular surveillance to monitor metallo-β-lactamase emergence.

The reviewed studies show notable diversity in the resistance patterns of MDR-*A. baumannii* to different antibiotics. When treating MDR-*A. baumannii* infections, carbapenem resistance—which includes resistance to imipenem and meropenem—is a serious problem. Moreover, the random-effects model estimate of 62.8% from the pooled analysis of carbapenem resistance indicated significant variability between trials, with substantial heterogeneity (I^2^ = 94.6%). These results point to a significant and pervasive problem with carbapenem resistance in Saudi Arabian ICU patients. Moreover, literature from other nations [51,52,53] reveals that in a research conducted in Jordan, tigecycline and colistin also had the lowest rates of *A. baumannii* resistance; still, 99% of isolates were carbapenem-resistant, and 76.8% were MDR [52]. Resistance to critical agents like imipenem and meropenem severely restricts treatment options, often leading to poorer patient outcomes and higher mortality in ICU settings. This underscores the need for urgent development of alternative therapies and strict infection control measures to curb the spread of resistant strains. We found that a significant heterogeneity was also seen in tigecycline resistance. Some investigations found a notably high level of gentamicin resistance. Hence, TMP-SMX may not be a dependable treatment choice for MDR-*A. baumannii* in many ICU settings, as shown by these significant resistance rates. Strengthening antimicrobial stewardship and considering alternative therapies may help address this variability in resistance.

The low incidence of colistin resistance, as shown by the random-effects model with a rate of 1.70%, is one of the more encouraging results from our research. Colistin is still recommended as a therapy for *A. baumannii* infections because of this low prevalence of resistance, a view that has been supported by more recent research by researchers [38,41]. These trials demonstrate the usefulness of colistin as an antibiotic used as a last option for serious infections. Colistin was the most often prescribed antibiotic in Malaysia, an Asian nation, for isolates of *A. baumannii* that were resistant to carbapenems [54]. Concerning findings, dissimilar from our study, were obtained from Egypt, where colistin resistance reached 20% [55]. Nonetheless, colistin resistance remained low despite these high resistance rates, suggesting colistin’s ongoing effectiveness as a therapeutic option. However, to preserve its effectiveness, colistin should be used judiciously within antimicrobial stewardship programs to prevent the emergence of resistance, as several countries have explored increasing resistance to colistin [56,57]. Our findings align with international patterns showing high resistance to carbapenems and preserved susceptibility to colistin. However, compared to other regional and global analyses, our pooled MDR prevalence (19.24%) and carbapenem resistance levels (>50%) are relatively lower. For instance, studies from Iran [58] and Ethiopia [59] reported carbapenem resistance exceeding 85% and MDR prevalence of 36.4%, respectively. Another meta-analysis identified key predictors for resistant *A. baumannii*, including prior carbapenem use, tracheostomy, mechanical ventilation, and ICU stay [60]. These risk factors closely mirror our ICU-specific findings, supporting the importance of localized infection control and antibiotic stewardship.

Because *A. baumannii* has a high level of antibiotic resistance, treating its infections in intensive care units may be difficult. Our review demonstrates that the majority of current treatment approaches depend on a small number of potent antibiotics, most notably colistin, which is still important because of its low resistance rates, even with its potential nephrotoxic effects, as demonstrated by studies [30,41]. Tigecycline is also involved; however, research by Alhaddad et al. and Al-Omari et al. indicates that its effectiveness is declining with increased resistance [28,38]. Combination treatments, such as colistin plus carbapenems or tigecycline, provide a viable strategy against strains of bacteria that are resistant to many drugs [60,61]. According to a prior study, several antibiotics used together may have a synergistic effect against *A. baumannii* [62]. More than 70% of isolates with comparatively low toxicity showed significant synergistic effects for colistin-glycopeptide and polymyxin-carbapenem combinations. Combination treatments greatly increased bactericidal activity as compared to monotherapy. Moreover, the colistin plus meropenem combination was better for the complete bacteriological response than colistin monotherapy [63,64]. Finally, the wide 95% prediction intervals observed across pooled models indicate that the true prevalence and resistance rates of MDR-*A. baumannii* may differ substantially among ICU settings in Saudi Arabia, reflecting variations in infection-control practices and antimicrobial stewardship intensity.

## 5. Strengths and Limitations

The current research provides an integrated view of *A. baumannii* data with respect to prevalence and resistance in Saudi Arabian ICUs, thereby contributing towards enhancing the region-based research that has been lacking in this regard. Further, the meta-analytical method adds to the reliability of the findings, especially with regard to the trends of resistance to antibiotics, such as those including the use of carbapenems and tigecycline, which are critical in coming up with practical management plans for the ailment. However, some limitations of this systematic review and meta-analysis should be highlighted. The large variations in the included studies in design, sample size, and methodology might also affect the generalizability of the findings. The observed heterogeneity in frequency and resistance rates may also be due to differences in patients, ICU characteristics, study duration, and locations within the country, more specifically, the geographical areas of Saudi Arabia. In addition, many researchers’ reliance on retrospective data creates a potential bias and diminishes the accuracy of the reported rates of prevalence and resistance.

## 6. Conclusions

The present study found a significant prevalence and different resistance profile of *A. baumannii* among ICU patients in Saudi Arabia. The findings indicate alarming rates of resistance levels to carbapenems and third-generation cephalosporins, and a wide array of prevalence rates to other antibiotics. So far, the increase in the rate of antibiotic resistance is worrisome. Yet, colistin remains the most viable alternative. The present study findings recommend the necessity for surveillance, judicious use of antibiotics, and appropriate infection control measures in patients admitted to the ICUs in order to control the effects of MDR-*A. baumannii*. Furthermore, investigation of the resistance mechanisms, as well as the influence of infection control policies on MRDR-*A. baumannii* transmission in the ICUs, is crucial for impacting clinical practice and health policy. A concerted effort is needed to address this public health issue by creating novel antimicrobial drugs, enhancing diagnostic tools, and putting evidence-based recommendations for MDR-*A. baumannii* infection prevention and treatment into practice. Future studies should aim at the use of standardized approaches and the conduct of large, multi-center, prospective studies to provide more robust and generalizable findings.

## Figures and Tables

**Figure 1 antibiotics-14-01131-f001:**
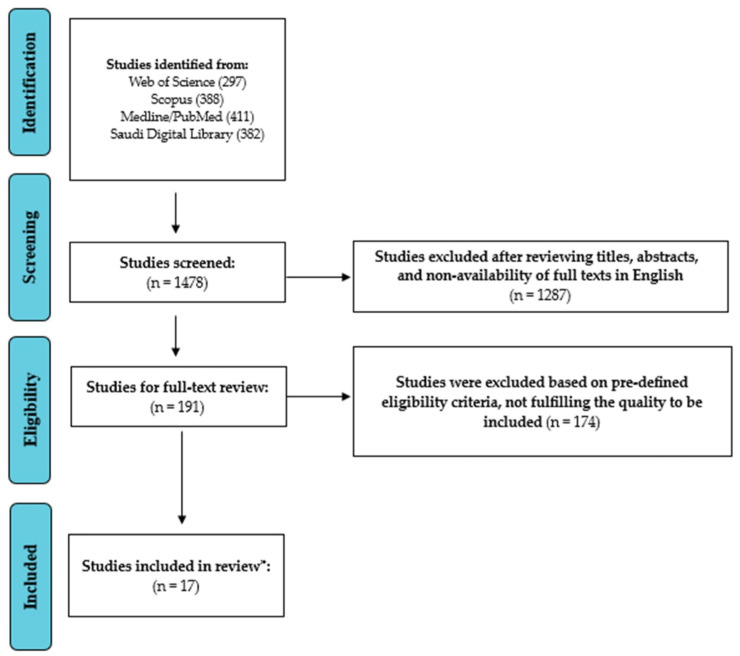
Study Screening and Selection Process (PRISMA Flow chart). * Number of studies included in the review.

**Figure 2 antibiotics-14-01131-f002:**
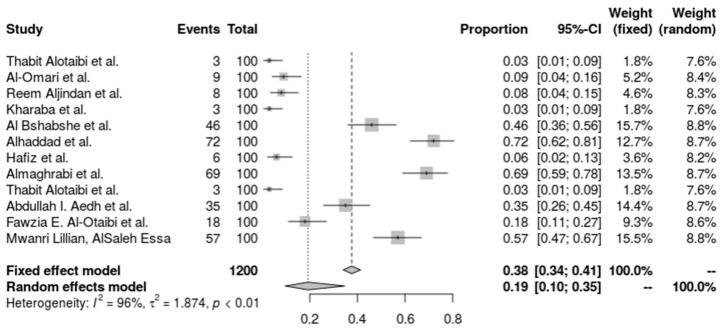
Forest plot of the prevalence of *A. baumannii.* Heterogeneity: I^2^ = 96% (95% CI 92–98%), τ^2^ = 1.874, Q = 272.25, *p* < 0.001 [27,28,29,30,31,36,37,38,40,41,42].

**Figure 3 antibiotics-14-01131-f003:**
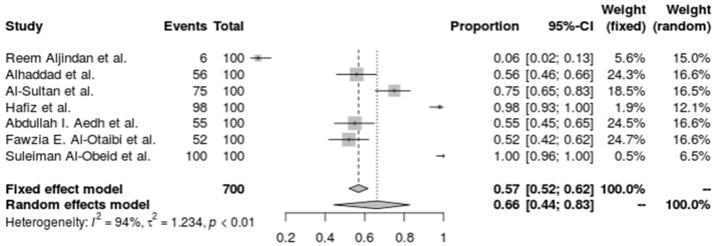
Forest plot of the resistance profile of *A. baumannii* against carbapenems. Heterogeneity: I^2^ = 94.6% (95% CI 90–97%), τ^2^ = 1.1998, Q = 110.28, *p* < 0.001 [30,31,32,36,38,39,41].

**Figure 4 antibiotics-14-01131-f004:**
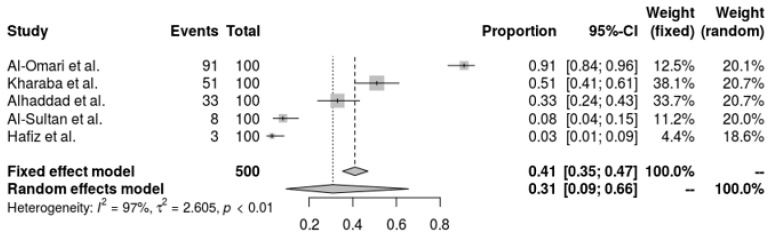
Forest plot of the resistance profile of *A. baumannii* against tigecycline. Heterogeneity: I^2^ = 96.8% (95% CI 92–98%), τ^2^ = 2.6051, Q = 125.39, *p* < 0.001 [28,30,37,38,39].

**Figure 5 antibiotics-14-01131-f005:**
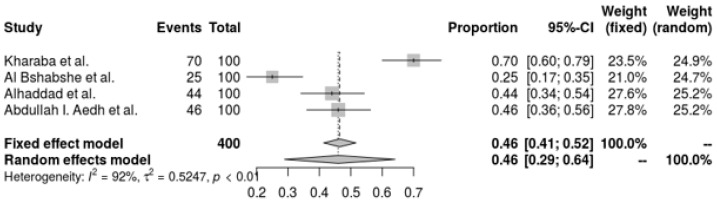
Forest plot of the resistance profile of *A. baumannii* Trimethoprim-Sulfamethoxazole. Heterogeneity: I^2^ = 95% (95% CI: 90–97%), τ^2^ = 1.453, Q = 98.67, *p* < 0.001 [29,37,38,41].

**Figure 6 antibiotics-14-01131-f006:**
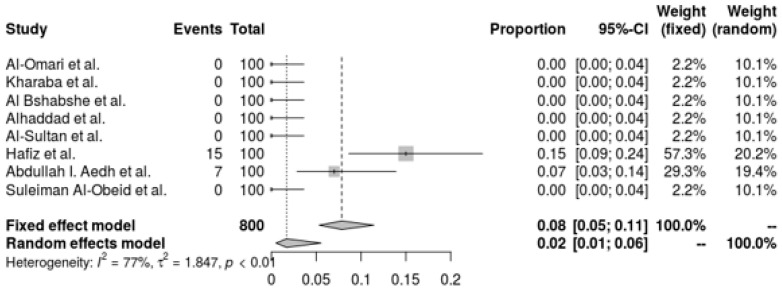
Forest plot of resistance profile of *A. baumannii* Colistin. Heterogeneity: I^2^ = 93% (95% CI: 88–96%), τ^2^ = 0.928, Q = 84.52, *p* < 0.001 [28,29,30,32,37,38,39,41].

**Table 1 antibiotics-14-01131-t001:** Inclusion and exclusion criteria applied in this systematic review and meta-analysis.

Criteria Type	Inclusion Criteria	Exclusion Criteria
Study Design	Observational studies (retrospective, prospective, cross-sectional, surveillance studies) reporting prevalence and/or antimicrobial resistance of *A. baumannii* in ICU patients in Saudi Arabia.	Reviews, case reports, editorials, conference abstracts, letters, experimental animal studies, in vitro-only studies.
Population	Adult or pediatric patients admitted to ICUs in hospitals in Saudi Arabia.	Studies not specific to ICU settings, studies outside Saudi Arabia, or studies without a clear patient population.
Outcomes	Studies reporting the prevalence of *A. baumannii* infections/colonization and/or resistance profiles to antimicrobial agents.	Studies not reporting prevalence or antimicrobial susceptibility data for *A. baumannii.*
Setting	Hospital-based, ICU-focused studies from Saudi Arabia.	Community-based studies, outpatient studies, or studies lacking a clear ICU context.
Language	English-language publications.	Non-English publications.
Timeframe	January 2014–30 September 2025.	Studies published before 2014 or after the last search date.

**Table 2 antibiotics-14-01131-t002:** Characteristics of the studies reviewed with prevalence and resistance profile.

Author	Year	Study Design	Population Type and Number	Mean Age	Prevalence of *A. baumannii*	Resistant Profile Against Antibiotics	Treatment
Mwanri L & AlSaleh E [42]	2014	Surveillance System Analysis	ICU patients, 496 HAI cases, 758 organisms identified	53	57.4% of MDROs	High prevalence of multidrug-resistant Acinetobacter species	Not specifically provided; highlights the burden of MDROs in ICU setting
Al-Obeid et al. [32]	2015	Retrospective	ICU patients, 506 (2006), 510 (2009), 936 (2012)	Not provided	Not provided	12 XDR-*A. baumannii* strains, all are resistant to meropenem and imipenem; all isolates are susceptible to colistin	Colistin and tigecycline combination showing good synergistic effect
Aljindan et al. [36]	2015	Prospective	ICU patients (565 rectal swab specimens)	Not specified	8.3%	Carbapenem-resistant (6.2%), OXA-23 β-lactamase main mechanism	Not specified
Al-Otaibi et al. [31]	2016	Retrospective	Oncology patients, 61 episodes in 56 patients	Not provided	18%	Imipenem AND Meropenem resistance 52.4%	Antimicrobial treatment based on susceptibility testing
Al Bshabshe et al. [29]	2016	Retrospective	ICU patients, 105	53.43	46.67%	100% sensitive to colistin, 74.5% to trimethoprim + sulfamethoxazole, 16.3% to amikacin, 7.7% to ampicillin, 7.3% to ceftazidime	Colistin, trimethoprim + sulfamethoxazole
Al-Omari et al. [28]	2017	Retrospective	Critically ill ICU patients (94 patients)	53.3 years	90%	Colistin: 100% Susceptible, 0% Resistant; Gentamicin: 33.33% Susceptible, 66.67% Resistant; Tigecycline: 8.33% Susceptible, 91.67% Resistant; Ampicillin/Sulbactam: 16.67% Susceptible, 83.33% Resistant	Colistin monotherapy vs. Colistin with Tigecycline, Meropenem, or both
Almaghrabi et al. [40]	2018	Prospective	Clinical isolates, 94	Not provided	69% of these isolates were multidrug-resistant strains	69% multidrug-resistant, high resistance to carbapenems, sensitive to one or two antibiotics	Not provided
Alhaddad et al. [38]	2018	Prospective	ICU patients, 11	Not provided	72.73%	56% resistance to imipenem and meropenem; 100% sensitivity to colistin; 12.5% resistance and 33.33% intermediate to tigecycline; 67% MDR; highest resistance to cephalosporins (78%) and fluoroquinolones (67%); 56% resistance to carbapenems and penicillin; 44% resistance to aminoglycosides and trimethoprim/sulfamethoxazole.	Not provided
Kharaba et al. [37]	2021	Prospective	ICU patients, 3179	57.28	3.9%	Antibiotic resistance: >80% for 12 of 16 antibiotics; Gentamicin (70.5%), Trimethoprim-sulfamethoxazole (69.8%), Tigecycline (51.6%), Colistin <50%	Comprehensive program for developing a local antibiogram database coupled with nationwide antimicrobial stewardship
Alotaibi et al. [27]	2021	Retrospective	ICU patients at King Fahad University Hospital (198 patients)	49 years (males), 56 years (females)	3.37%	Higher resistance in non-survivors; colistin and carbapenem resistance observed	Colistin (65 patients), Carbapenems (18 patients), Combination (22 patients)
Al-Sultan [39]	2021	Prospective	ICU patients in Makkah and Al-Madinah, number not provided	Not provided	Not provided	In Al-Madinah, 84% resistance to imipenem and 88% to meropenem, with 8% resistance to tigecycline; in Makkah, 75% resistance to both imipenem and meropenem, with 100% sensitivity to colistin and tigecycline;	Not provided
Hafiz et al. [30]	2023	Retrospective	Mechanically ventilated adult ICU patients, 115	74.5	6.2%	62% of isolates were MDR, with 99% resistant to piperacillin/tazobactam, 98% resistant to carbapenems, and 100% resistant to amoxicillin/clavulanate, cephalosporins, and fluoroquinolones. Colistin resistance was 15%, tigecycline resistance 3%. Sensitivity: 60% to aminoglycosides, 85% to tigecycline and colistin.	Colistin, tigecycline, emphasis on appropriate antibiotic therapy including these drugs during COVID-19 pandemic
Aedh et al. [41]	2023	Prospective	ICU patients, number not provided	males was 48.4 and females was 42.8	35.44%	Ceftazidime 89.2%, cefotaxime 78.5%, cefepime 71.4%, piperacillin/tazobactam 67.8%, imipenem 53.5%, meropenem 57.1%, ciprofloxacin 78.5%, levofloxacin 75%, amikacin 67.8%, tobramycin 57.1%, gentamicin 14.2%, trimethoprim-sulfamethoxazole 46.4%, colistin 7.1%.	Gentamicin, colistin, combination of colistin and trimethoprim/sulfamethoxazole
Gaifer et al. [35]	2024	Retrospective study	Hospitalized patients n = 138 (ICU cases, 87%)	69 years (mean)	Not separately reported for ICU	83% carbapenem-R; 87% ceftazidime-R, 85% piperacillin/taz-R, 85% ciprofloxacin-R; 89% colistin-S, 55% tigecycline-S	CRA cases mainly treated with colistin and tigecycline
Kaki R [33]	2024	Retrospective (chart review)	Patients with *A. baumannii* bacteremia, n = 112 (65 ICU cases: 54.5% MICU, 3.6% SICU)	58 years	Not applicable (focus on bacteremia); ICU admissions were 58% of cases	High resistance: carbapenems ~68% R, piperacillin/taz 73%, cephalosporins ~70%, fluoroquinolones ~70%, gentamicin 51%, TMP/SMX 66%; low resistance: colistin 4.5%, tigecycline 1.8%, amikacin 1.8%	Most common regimens: colistin + meropenem (56%), meropenem alone (19%)
Alharbi et al. [34]	2025	Retrospective (observational)	Tertiary-care hospital inpatients (2013–2023); ICU subset: 2934 *A. baumannii* isolates	Not reported	ICU accounted for ~51% of *A. baumannii* isolates	Very high: β-lactams 88–100% R, aminoglycosides 31–95% R, quinolones 43–99% R; colistin 3–42% R	Not specified in ICU context
Alanazi FA et al. [43]	2025	Laboratory-based cross-sectional molecular characterization of clinical isolate	ICU patients, Riyadh (King Saud University Medical City); *A. baumannii* isolates n ≈ 67	Not reported	Not applicable (isolate-based study; prevalence among patients not estimated)	Extremely high phenotypic resistance by VITEK-2: ampicillin/sulbactam 95.5%, piperacillin/tazobactam 95.5%, cefepime 95.5%, ceftazidime 94%, cefotaxime 94%, ciprofloxacin 94%, meropenem 94%, imipenem 94%; trimethoprim/sulfamethoxazole 93%; gentamicin 90%. Colistin: broth-microdilution subset (n = 11) showed 27% (3/11) resistance. Detected resistance genes	No therapeutic evaluation performed; study focused on molecular and phenotypic resistance analysis

## Data Availability

This systematic review and meta-analysis are based on data extracted from previously published studies, all of which are cited in the manuscript. No new raw data were generated.

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
