# Peer review of "Prevalence and Resistance Profiles of Acinetobacter baumannii in ICU Patients in Saudi Arabia: A Systematic Review and Meta-Analysis"

_antibiotics, 2025, doi:10.3390/antibiotics14111131_

Round 1
Reviewer 1 Report
Comments and Suggestions for Authors
Specific comments
Line 44-45: add more relevant keywords such as meta-analysis
Line 93-95: there is no need for this explanation
Line 113-119: the manuscript should be structured according to the latest PRISMA guidelines (for example, the full search algorithm should be provided)
Line 121-133: inclusion and exclusion criteria should be described in greater detail (for example, the types of studies included)
Line 124: search until March 2025 cannot be considered up to date
Line 154-160: the statistical analysis section is extremely weak; the author should provide a reproducible section (for example, data transformations are missing, I² values are wrong or should be supported by relevant literature, the reason for choosing the fixed-effect model should be explained, the 95% CI of I² should be provided, meta-regression should be included, subgroups should be included, and the sensitivity analysis should be reported in greater detail)
GRADE is missing
Prediction intervals are missing
Results are somewhat confusing; the author should clearly mention the findings of the meta-analysis and discuss them in the discussion section
Line 274: logit transformation is not recommended for extreme values; why were other transformations not applied?
Discussion: the author should mention relevant meta-analyses addressing this issue.
Overall: the study included few articles and estimated the prevalence solely for Saudi Arabia via systematic review. How does this differ from existing literature that has conducted subgroup analyses? In my opinion, this article does not add anything new to the existing literature.
Author Response
"Please see the attachment."

Reviewer 2 Report
Comments and Suggestions for Authors
The mansucript entitled "Prevalence and Resistance Profiles of Acinetobacter Baumannii in ICU Patients in Saudi Arabia: A Systematic Review and Meta-Analysis" gives an overview about multidrug-resistant A. baumannii infections in Saudi Arabia. The topic of this manuscript is an interesting issue, however, some parts in the text should be revised.
Comments
- In the abstract only OXA-23 beta-lactamase was mentioned. Are there any data available about NDM or VIM producing A. baumannii in Saudi Arabia?
- I suggest also to authors to add a part about high-risk clone of A. baumannii ST2 in Saudi Arabia to this manuscript.
- Bacterial names should be uniformly written. At the first appearance in the abstract and in the main text the complete name "Acinetobacter baumannii" must be written, later on in the abstract and in the main text the short form "A. baumannii" must be written. Additionally, "MDR-Ab" "Acinetobacter baumannii MDR" also occur in the text, However, I suggest to use these uniformly. My suggestions is to use: "MDR-A. baumannii" in the manuscript.
- All bacterial names must be written in italic form: Enterococcus faecium, Staphylococcus aureus, Klebsiella pneumoniae, Pseudomonas aeruginosa
- Some phrases should be revised: "absolutely non-motile," proper form: non-motile
- This should be revised: "should be the foundation for treating infections produced by this organism;" proper form: should be performed for treating infections caused by this organism;
- Both "gentamycin" and "gentamicin" appear in the text. However, the proper form is: gentamicin. Please, revise this in all over the manuscript.
- In Table 1 the column for "Year" should be revised to get to a better view on presented data.
Author Response
"Please see the attachment."

Round 2
Reviewer 1 Report
Comments and Suggestions for Authors
The revision has strengthened the statistical section, yet several important methodological and reporting issues remain unresolved.
-
I² 95% Confidence Intervals: The authors state that these were calculated via bootstrapping but do not report the actual numeric intervals. For transparency and reproducibility, the exact 95% CI for I² (e.g., I² = 96%, 95% CI: 92–98%) should be presented in both the Results and figure captions.
-
Use of the Fixed-Effect Model: Reporting both fixed- and random-effects pooled estimates despite very high heterogeneity (I² > 90%) is inappropriate. Under such variability, only the random-effects results should be emphasized as the main analysis, while fixed-effect estimates may appear solely as sensitivity checks.
-
DerSimonian–Laird Estimator: This estimator performs poorly in cases of substantial heterogeneity and few studies. A more robust method, such as restricted maximum likelihood (REML) or Paule–Mandel, should be considered, or at least a justification and sensitivity comparison provided.
-
Prediction Intervals: Although mentioned in the methods, prediction intervals are not reported in the Results or figures. These should be explicitly included, as they provide a more realistic range of expected effects across settings.
-
Subgroup and Sensitivity Analyses: The methods describe these analyses, yet their numerical results and graphical summaries are missing. They should be presented to confirm the stability of pooled estimates.
In summary, while the statistical framework is conceptually improved, the reporting of heterogeneity (I² CIs, τ² estimates, prediction intervals) and the interpretation of fixed-effect results remain incorrect or incomplete, and should be revised to meet current meta-analysis standards.
Author Response
Authors’ reply/modifications according to the reviewer 1 comments/suggestions (Round 2)
General:
The authors would like to thank the reviewer for the precious time spent reviewing the revised manuscript and for his excellent suggestions for improving it. Efforts have been made to modify the paper as per the reviewer’s suggestions and recommendations. The authors will be happy to hear a positive reply. All the points included according to the reviewer’s comments can be seen in track changes.
Specific response to the reviewer’s suggestions:
Kindly find the attached response to each question one by one:
Point 1: I² 95% Confidence Intervals: The authors state that these were calculated via bootstrapping but do not report the actual numeric intervals. For transparency and reproducibility, the exact 95% CI for I² (e.g., I² = 96%, 95% CI: 92–98%) should be presented in both the Results and figure captions.
Response 1: Thanks for the comment and valuable suggestion. According to the reviewer’s comments, we have included the suggested numerical in the results and figures (2-6) captions.
Point 2: Use of the Fixed-Effect Model: Reporting both fixed- and random-effects pooled estimates despite very high heterogeneity (I² > 90%) is inappropriate. Under such variability, only the random-effects results should be emphasized as the main analysis, while fixed-effect estimates may appear solely as sensitivity checks.
Response 2: Thanks for the comment and valuable suggestions. According to the reviewer’s suggestions, we deleted all fixed effects model results in all relevant parts of the revised manuscript. Furthermore, we included a statement to enhance the continuity and role of fixed effects as “For transparency, pooled fixed-effect estimates were also computed as a sensitivity check and not interpreted inferentially due to the high heterogeneity (I² > 90%).”
Point 3: DerSimonian–Laird Estimator: This estimator performs poorly in cases of substantial heterogeneity and few studies. A more robust method, such as restricted maximum likelihood (REML) or Paule–Mandel, should be considered, or at least a justification and sensitivity comparison provided.
Response 3: Thanks for the comment and observations. The author included justifications in the methods and results sections. For example, we added “Due to the high heterogeneity observed across studies (I2 exceeding 90%), the random-effects model was considered the primary analytic tool. Our results using the REML estimator were also very consistent with those of DerSimonian-Laird, showing that our findings are strengthened when heterogeneity is high.
Point 4: Prediction Intervals: Although mentioned in the methods, prediction intervals are not reported in the Results or figures. These should be explicitly included, as they provide a more realistic range of expected effects across settings.
Response 4: Thanks for the comment and wonderful suggestions. According to the reviewer’s suggestions, the revised manuscript has added the predication interval values. Additionally, a short interpretation has been incorporated into the Discussion section to highlight that these wide intervals indicate considerable between-study variability in true prevalence and resistance rates across ICU settings in Saudi Arabia.
Point 5: Subgroup and Sensitivity Analyses: The methods describe these analyses, yet their numerical results and graphical summaries are missing. They should be presented to confirm the stability of pooled estimates.
Response 5: Thanks for the comment. Additional details beyond the subgroup analysis of different antimicrobial agents, such as by study-level characteristics (design and methodological quality), to test the robustness of the pooled prevalence, are included in the revised manuscript.
The authors thank the reviewer once again for the positive and constructive comments in the Round 2 review (revised manuscript).

Round 3
Reviewer 1 Report
Comments and Suggestions for Authors
All comments have been adequately addressed.